

# Biodiversity data obsolescence and land uses changes

Nora Escribano, Arturo H. Ariño and David Galicia

Department of Environmental Biology, Universidad de Navarra, Pamplona, Spain

## ABSTRACT

**Background**. Primary biodiversity records (PBR) are essential in many areas of scientific research as they document the biodiversity through time and space. However, concerns about PBR quality and fitness-for-use have grown, especially as derived from taxonomical, geographical and sampling effort biases. Nonetheless, the temporal bias stemming from data ageing has received less attention. We examine the effect of changes in land use in the information currentness, and therefore data obsolescence, in biodiversity databases.

**Methods**. We created maps of land use changes for three periods (1956–1985, 1985–2000 and 2000–2012) at 5-kilometres resolution. For each cell we calculated the percentage of land use change within each period. We then overlaid distribution data about small mammals, and classified each data as 'non-obsolete or 'obsolete,' depending on both the amount of land use changes in the cell, and whether changes occurred at or after the data sampling's date.

**Results**. A total of 14,528 records out of the initial 59,677 turned out to be non-obsolete after taking into account the changes in the land uses in Navarra. These obsolete data existed in 115 of the 156 cells analysed. Furthermore, more than one half of the remaining cells holding non-obsolete records had not been visited at least for the last fifteen years.

**Conclusion**. Land use changes challenge the actual information obtainable from biodiversity datasets and therefore its potential uses. With the passage of time, one can expect a steady increase in the availability and use of biological records—but not without them becoming older and likely to be obsolete by land uses changes. Therefore, it becomes necessary to assess records' obsolescence, as it may jeopardize the knowledge and perception of biodiversity patterns.

Corresponding author
Nora Escribano,
nescribano@alumni.unav.es

## INTRODUCTION

Primary biodiversity records (PBR) are essential in biogeography, trend assessment, climate change ecology or conservation biology (*Rocchini et al., 2011*; *Powney & Isaac, 2015*). At their most basic level, PBR provide fundamental data about location of living beings—what, where and when—(*Ariño et al., 2012*) and have been widely used for building maps and models of species distribution.

Interest in this type of data, which provides direct information about where organisms have been previously reported, has grown in recent years as the loss of biodiversity has

become one of the main concerns in the scientific community (*Hermoso, Kennard & Linke, 2013*). The relevance and utility of these data are both important for assessing priority areas for conservation and determining drivers of biodiversity loss (*Willis et al., 2007*; *Powney & Isaac, 2015*, *Convention on Biological Diversity (CBD), 2010*).

Free access to PBR has been steadily increasing in the last decade as many initiatives of data sharing have been launched (e.g., GBIF, VerNet, Living Planet Index). However, concerns about the quality and fitness-for-use of this data have grown since massive biodiversity data have become available (*Hill et al., 2010*; *Ariño, Chavan & Fatih, 2013*; *Faith et al., 2013*). The quality of data responds to different sources of errors and uncertainties, and these determine their usability. Typically, information in datasets is spatially and taxonomically biased and differs in the amount of sampling effort (*Boakes et al., 2010*) so researches have focused on addressing which roles play these limitations on data quality (*Sastre & Lobo, 2009*; *Otegui et al., 2013*; *Beck et al., 2014*). All of this has led to important research articles and guidelines about how to avoid and reduce data uncertainty (e.g., *Chapman, 2005*).

Nevertheless, researchers have paid less attention to temporal bias. By definition, a PBR is always a piece of information about the past (where and when an organism *was* recorded). Thus, its usefulness for estimating current ecological phenomena (like distribution or niche preference) will be affected by its age. As time goes by and the PBR becomes older, uncertainty about its spatial information is expected to increase. Obviously, the population may have gone extinct locally and thus, we cannot assure that, at the present, the PBR is informative whether the population still spreads there or not. However, it is also worth noticing that the habitat present at the locality may have changed since the species was recorded and thus, any matching between specimen location and habitat conditions must assure the temporal synchrony of both aspects of the species' ecology. Comparatively, while information about locations of individuals in the past is abundant, information about past conditions of the territory (e.g., land uses) is not as common. As this temporal gap grows, we cannot guarantee that the current environmental conditions resemble those when the observation or specie was recorded. We may thus expect this uncertainty to eventually reach some threshold value that would turn it into an outdated PBR, unsuitable for being used in models under current habitat conditions. In this way, we use the term obsolete to define a PBR whose temporal frame no longer supports the relationship between the referred specimen and the current habitat conditions. In many cases, the uncertainty of older records has led to include only recent ones in studies about distribution patterns or niche modelling (*Moscoso, Albernaz & Salomão, 2013*; *Ficetola et al., 2014*). However as far as we know, the expiration of the records has not been assessed in literature.

At short (e.g., intra-successional) time scales, communities tend to show a characteristic composition as long as habitat conditions remain stable. Their persistence will vary according to multiple factors, like their ecological structure and species' natural history (*Pimm, 1991*), but ecosystem alterations may induce permanent changes in communities. Thus, the uncertainty about the presence of some species will be higher in ecosystems that have undergone modifications between the time of record and the time of record's usage. Also, as land use changes, biological invasions and other causes of rapid diversity

loss take place (*Sala et al., 2000*; *Crowl et al., 2008*), obsolescence could become even more of a crucial factor. Humans have a large trajectory of changing landscapes, being one of the most highlighted drivers of biodiversity loss (*Ellis et al., 2010*). Recent studies have shown the direct relation between the changes in the use of land and changes in bird (*Rittenhouse et al., 2012*; *Dorresteijn et al., 2015*), mammal (*Sieber et al., 2015*; *Torre et al., 2015*; *Cisneros, Fagan & Willig, 2015*), amphibian (*Nori et al., 2015*) and invertebrate communities (*Ngai et al., 2008*; *Wagner, Krauss & Steffan-Dewenter, 2013*). Therefore, when the dates of records in a biodiversity database coincide with, or precede, important changes in land use, neglecting obsolescence of the data could allow for a considerable amount of noise in the primary data used to estimate potential distribution areas or diversity patterns.

In this paper we investigate the effect of changes in land use on the obsolescence of data contained in biodiversity databases. For this purpose, we need long term biodiversity data from an area where changes in the territory have occurred, but also that this geographical information is accessible or even exists. These two requirements are met in the Vertebrate Collection of the Museum of Zoology of University of Navarra (hereafter MZNA) which holds mammal's long term data from the state province of Navarra. Moreover, the geographic information system of Navarra (SITNA) gathers land uses information covering the temporal data frame of the MZNA dataset. Thus, this study extent together with the rich data from mammals in this area makes it a valuable testing ground for the target of this research.

## MATERIALS AND METHODS

### Study region
Navarra is a 10,391 km square region located in the north of Spain, wedged between the western end of the Pyrenean Range and the Ebro Basin (Fig. 1). Both Eurosiberian and Mediterranean bioregions occur in this territory. It gathers a high landscape diversity as a result of its wide variety of climatic and topographic conditions (*Nogués-Bravo, 2006*). The northern areas, with mean altitudes above 800 m (range 200 m–2,000 m), high precipitations (1,200–1,700 mm of mean annual rain and snowfall) and moderate temperatures, are covered by pastures and natural beech and oak forests as well as pine plantations. Agricultural and Mediterranean landscapes dominate the south of Navarra, with mean altitudes below 400 m and a combination of low precipitation (below 400 mm) and temperature that favour the presence of rich communities of xerophytic shrublands. All this variety of habitats determines the high diversity of the small mammals' community in Navarra, which comprises 31 of the 34 species of native insectivorous and rodents cited in the Iberian Peninsula (*Escala et al., 1997*).

### Biodiversity data
Data about small mammals for the region were extracted from the Pellet sampling dataset (*Escribano et al., 2016*) from the Vertebrate Collection of the Museum of Zoology of the University of Navarra. This dataset includes information of small mammals' records obtained from the analysis of barn owl's pellets and comprises 75% of the Collection in a time range spanning from 1967 to 2011. At this moment, the dataset constitutes the best

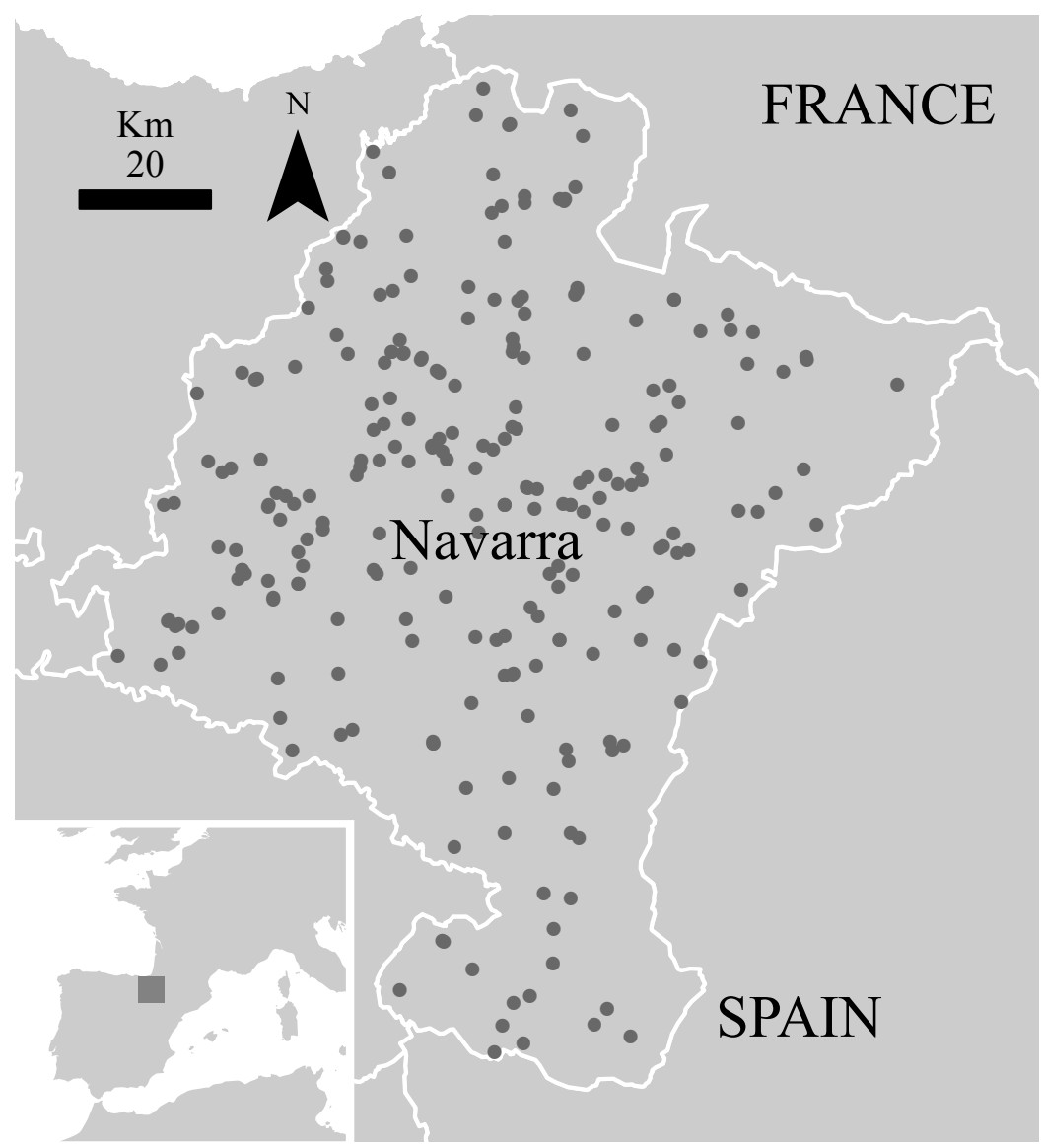

**Figure 1** **Sampling sites for small mammals in the 'Pellet sampling' dataset in Navarra.** Inset: location of Navarra within the Iberian Peninsula. Outset: Navarra in white line.

comprehensive, publicly accessible collection of small mammals in Spain (Figs. 1 and 2A). As pellets have proved to be very useful for assessing the small mammals' community in a territory (*Torre, Arrizabalaga & Flaquer, 2004*; *Avenant, 2005*), this dataset is a valuable tool for analysing the changes in distribution patterns of diversity in the last four decades.

In order to avoid overestimating the abundance of individuals, only records from skull remains where selected for analyses. Records from species rarely predated by barn owl or records lacking in coordinates and date of the sampling event were also discarded. In total, 59,677 records of 27 species of small mammals were used for the study (see Table S1).

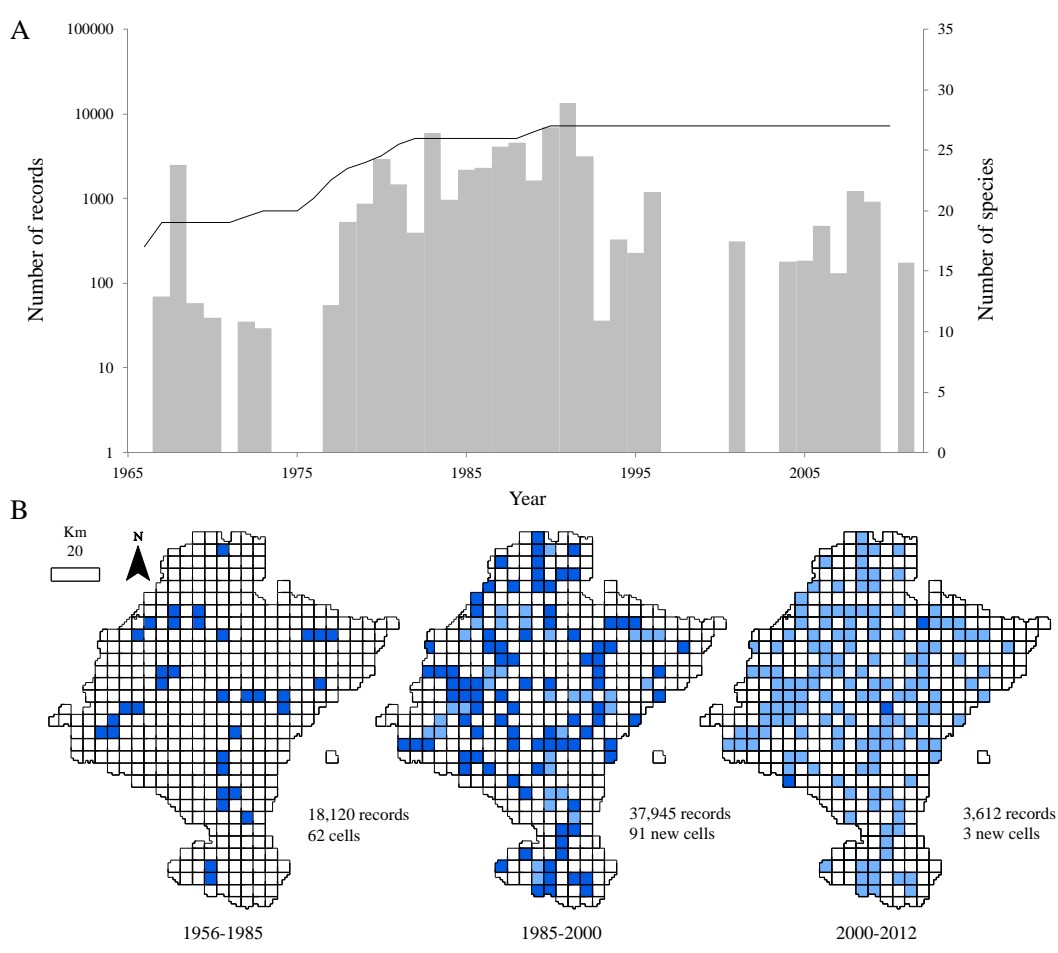

**Figure 2** **Temporal and spatial distribution of records in 'pellet sampling' dataset.** (A) Temporal distribution of the records. Bars: historical registry of small mammals in Navarra (as number of records per year). Line: species accumulation curve. (B) Distribution of records according to the three time intervals. Dark blue: cells first sampled in the time interval considered. Light blue: cells already sampled before the interval.

## Land use data

Land use data was obtained from the Geographic Information System of Navarra. For our purpose, we chose the land use maps for 1956, 1985, 2000 and 2012 covering the entire time span in our occurrence data. All maps are polygon-based layers that provide information about land cover types in Navarra.

We reclassified the land uses categories into five new categories attending to the structure of the vegetation cover: forest, bushes and meadows, herbaceous crops, woody crops and unproductive areas. In order to standardize the polygon's information and allow the later comparison of the different maps, each polygon was summarized by a relative abundance matrix of the five new categories considered. In this way, homogeneous polygons were expressed as matrices with a value of 1 in the appropriate category and 0 in the rest, while mixed polygons showed the proportional values for each category (that together equal the unit). For further details on the standardization of the maps see Document S1.

## Land use changes and biological data analysis

We used ArcGIS 10 software to perform spatial analyses (*ESRI, 2015*). We calculated the land uses changes for the three periods delimited by the reference maps (1956–1985, 1985–2000 and 2000–2012) by means of a spatial intersection of the corresponding maps. The three resultant maps of land use changes consisted on a series of polygons that either (1) had not suffered from changes in their land uses (there were no changes in their relative abundance matrices) or (2) had change from one land use to another (their relative abundance matrices were different). A 5 × 5 km grid was then superimposed to the study area, summarizing the percentage of land use changes per cell by dissolving all their polygons (see Document S1). The spatial resolution of the study (25 km sq) was set according to the assumed home range of the barn owl as the mammals' records come from pellets of this bird. A radius of 3 km from the nest/roosting site (and then an ideal circular area of about 28 km sq) is accepted to be suitable to characterize the habitat of this predator in a territory (*Martínez & Zuberogoitia, 2004*). The small mammals' records were also collapsed to the same grid. Cells with no recorded occurrences as well as those with an effective size below 12.5 km sq (mostly located in the border of the study area) were discarded.

In order to evaluate the records' obsolescence, cells with more of the 25% of their surface affected by land use modifications were labelled as highly modified cell (HMC). The threshold was empirically established based on those cells with sampling series long enough to perform comparative analyses of community and territorial changes (see Document S1 for further details). Then, following the reasoning presented previously, samples collected before or during periods of land use changes were labelled as obsolete.

## RESULTS

### Temporal variation in data sources

Our analysis of the net changes in the land uses of Navarra showed that 27% of its surface has changed during the entire dataset's timeframe (1956–2012). Overall, forestry land uses increased by 11% while meadows and shrublands decreased by 12%. Herbaceous and woody crops suffered a small decrease, and unproductive areas increased slightly. Relative to HMC, we found that they were greater in the periods 1956–1985 and 1985–2000, representing more than the 60% of cells. On the contrary, between 2000 and 2012 the number of HMCs decreased and there were only 14 HMC.

After discarding records from species rarely preyed upon by the barn owl and records that did not come from the study area, the dataset provided 59,677 georeferenced records of 27 species of small mammals. Despite the relatively large number of records in a relatively small area, little more than one third of the cells held data (36.02%, 156 cells). Also, the temporal distribution of records from 1966 to 2011 showed an evidently clumped pattern, with a main sampling period centred on the 80's (Fig. 2A). This period corresponds with the years leading to the publication of a regional atlas about the insectivorous and rodents community (*Escala et al., 1997*). Although most of the sampling effort was made between 1987 and 1992 (more than 55% of the records come from those years), we can

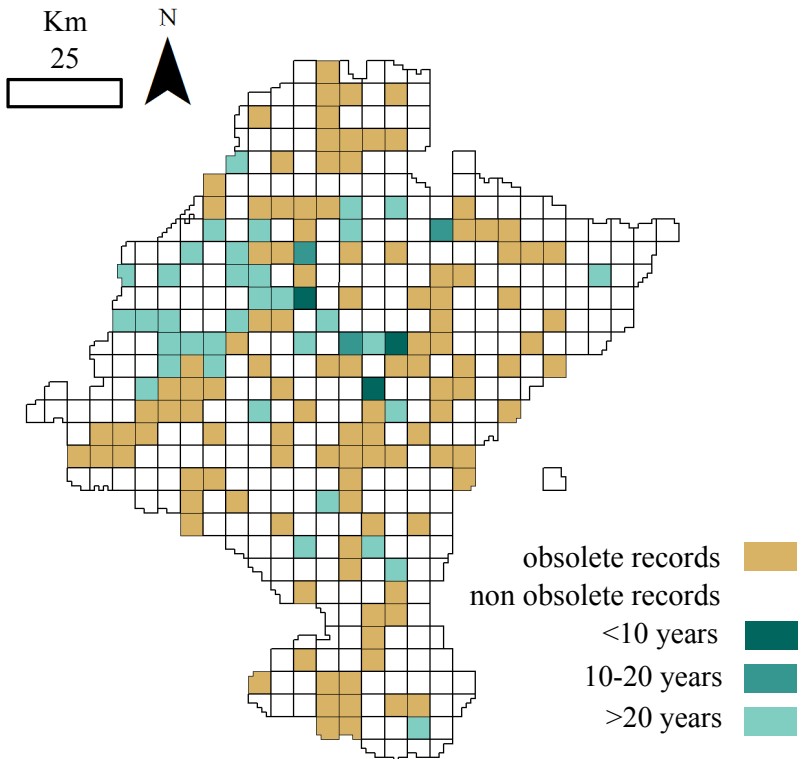

**Figure 3** **Distribution of cells holding small mammals' records.** White: cells with no data; brown: cells with obsolete data; blue: cells with non-obsolete data. Blue shades: length of time since the cell was last visited (darker: more recent visit).

observe another period where records from pellets increased between 1980 and 1987. This inventorying was not large in terms of occurrences, but was remarkably uniform through time. After the publication of the atlas, addition of new records of small mammals decreased.

Regarding to the richness of the dataset, all the 27 species considered in this study were already present in the database by 1991 and richness had almost attained a plateau five years before (Fig. 2A). Even though samples collected after 1985 did not contribute significantly to the total richness of the group, the spatial representativeness of the dataset grew considerably (Fig. 2B). The sampled area duplicated from 1985 to 2000, adding 91 new cells, previously unsampled, and thus increasing the biogeographical information of most of the species.

## Obsolescence analysis

A total of 115 of the 156 cells analysed (73.72%) contained obsolete data, that is, some of the samplings were performed before or during periods when the land use changed. If we exclude these outdated occurrences, the dataset information fells down to 14,528 records belonging to 24 species. Although cells holding these non-obsolete records have not suffered from considerable land use changes (<25% of its surface changed) ever since samplings occurred, more than one-half of these cells were not visited again over the last fifteen years (Fig. 3).
**Table 1** **Overview of information contained in the dataset.** (a) All samples and (b) non obsolete records.

| Species | (a) Entire dataset | | (b) Non obsolete records | |
|---|---|---|---|---|
| | Records (count) | Cells (count) | Records (count) | Cells (count) |
| *Apodemus flavicollis* | 541 | 43 | 177 | 16 |
| *Apodemus sylvaticus* | 9,837 | 148 | 2,607 | 45 |
| *Arvicola sapidus* | 152 | 47 | 27 | 15 |
| *Arvicola terrestris* | 166 | 8 | 37 | 2 |
| *Crocidura russula* | 14,585 | 146 | 3,929 | 45 |
| *Crocidura suaveolens* | 33 | 8 | | |
| *Eliomys quercinus* | 27 | 14 | 1 | 1 |
| *Glis glis* | 4 | 3 | 3 | 2 |
| *Micromys minutus* | 62 | 13 | 16 | 4 |
| *Microtus agrestis* | 3,459 | 113 | 1,205 | 42 |
| *Microtus arvalis* | 491 | 12 | 86 | 3 |
| *Microtus cabrerae* | 2 | 1 | | |
| *Microtus duodecimcostatus* | 7,983 | 135 | 2,092 | 45 |
| *Microtus gerbei* | 1,353 | 89 | 484 | 37 |
| *Microtus lusitanicus* | 2,598 | 54 | 226 | 21 |
| *Mus domesticus* | 815 | 82 | 212 | 24 |
| *Mus spretus* | 11,734 | 123 | 2,238 | 39 |
| *Myodes glareolus* | 326 | 48 | 110 | 21 |
| *Neomys anomalus* | 30 | 19 | 12 | 8 |
| *Neomys fodiens* | 302 | 48 | 93 | 20 |
| *Rattus norvegicus* | 102 | 38 | 13 | 7 |
| *Rattus rattus* | 17 | 12 | 4 | 3 |
| *Sorex araneus* | 2 | 1 | | |
| *Sorex coronatus* | 3,573 | 94 | 789 | 37 |
| *Sorex minutus* | 552 | 54 | 88 | 25 |
| *Suncus etruscus* | 913 | 91 | 77 | 20 |
| *Talpa europaea* | 18 | 8 | 2 | 2 |
| | 59,677 | | 14,528 | |

Species by species, the overall attrition rate (number of records filtered out by flagging them as obsolete) was generally high, over 70% in 74% of the species. The information loss was critical in species with very low representation in the original dataset, such as *Crocidura suaveolens* Pallas, 1811, *Microtus cabrerae* (Thomas, 1906) and *Sorex araneus* Linnaeus, 1758. Although an overall reduction in record numbers is expected to affect all species to a greater or lesser extent, these species actually disappeared from the dataset (Table 1).

## DISCUSSION

### Biodiversity records in a changing territory

The land use analysis showed a generalized increase in forested surface along the considered time, while meadows and shrublands show the opposite tendency. These two changes occurred mainly in the earliest period (from 1956 to 1985). The reduction of meadow land

seemed related to an increase in forest cover due mainly to two managerial practices: (1) steady repopulation on bare soils with tree species as an erosion-avoiding strategy, and (2) phasing out of fuel in favour of other energy sources, which occurred mainly between 1970 and 1990 (*Gobierno de Navarra, 2000*; *Nogués-Bravo, 2006*). However, our study does not aim to understand how Navarra has shifted its landscape, but to record the main land use changes at the sites where small mammals were recorded. These changes might therefore compromise the information that could be obtained from the biodiversity datasets.

We found that more than half the record-bearing territory area contained records compromised by such changes, and thus rendering these records as potentially obsolete. After applying our criterion of filtering out PBR collected before or during land use changes, we discarded approximately 75% of the small mammals' records (Fig. 4). As reported in several studies, changes in the management of the territory have led to changes in communities and older records may not reflect the current communities. For example, *Torre et al. (2015)* resampled historically sampling sites for barn owl pellets finding changes in the frequencies of small mammals and a decrease in the species richness from 14 to 10. Similarly, when we discarded data compromised by land use changes, richness in our 'Pellet sampling' dataset lost three of the rarest species. Furthermore, the number of cells in which each species was recorded was reduced in most cases by more than 70%. The loss of information was not concentrated at specific sites, but spread all over the studied territory.

## Implications of data obsolescence

Obsolescence could become a critical factor for species whose distribution pattern remains unclear. For example, *M. cabrerae* is an endemism of the Iberian peninsula, but its distribution is still unclear as information about its presence is patchy (*Garrido-García et al., 2013*). This knowledge gap ushered this microtine into the UICN Red List as 'Nearly Threatened' (see Table S1) and catalogued as 'Vulnerable' in Spain (*Palomo, Gisbert & Blanco, 2007*). However, if the data records of this species are old enough and come from a territory where land use changes have occurred afterwards, we cannot longer guarantee that those records are contributing to a reliable, current picture of its actual distribution. Similarly, other species whose distribution is unknown will suffer from the same lack of confidence.

Obsolescence could also affect the outcomes of SDMs when the habitat descriptors do not match the data frame of the biodiversity records used to calibrate the model, thus giving a biased result. The main challenge is then to find a way of estimating how much this ageing in relation to a desired timeframe, e.g., the 'now,' may affect the outcome of the model. As PBR are central to the construction of these models, it is important to considerate this limitation as well as other types of biases.

On the contrary, by highlighting the obsolete information of a dataset we can identify which parts of the territory are not appropriately sampled according to the recent history of human interventions in the area. These areas can then be targeted for surveying. Due to the current budget limitations for inventorying and completing databases with new records, it is imperative to design suitable surveys (*Sánchez-Fernández et al., 2011*) and monitoring programs (*Boero et al., 2015*) that optimize the available resources into a significant increase

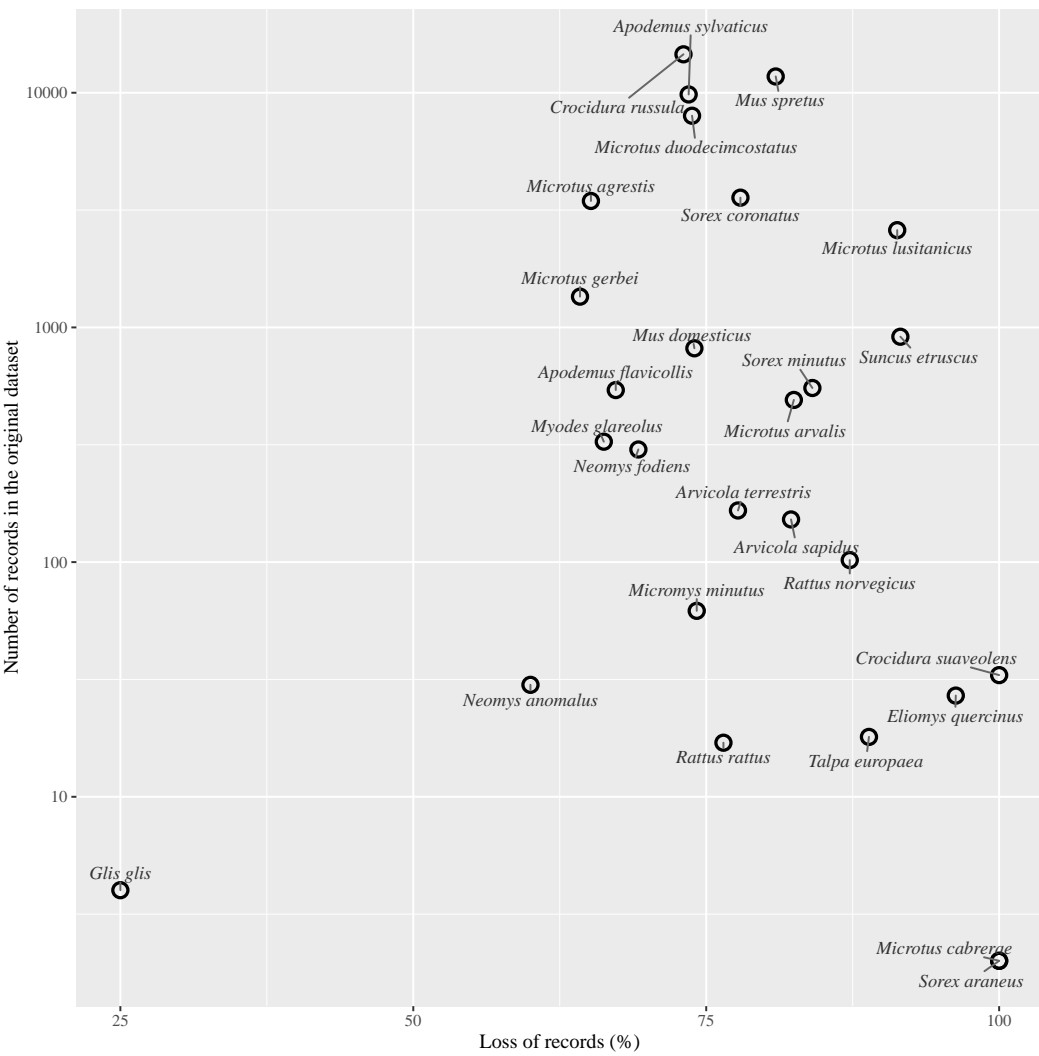

**Figure 4  Effect of records loss per species.** Relationship between the percentage of records loss by species and the number of records gathered in the entire dataset (log scale).

of biodiversity information. Our results showed that records from abundant species are prone to a lower decrease whereas records from rare species show a greater rate of attrition. Together, these two observations can help guide sampling efforts towards more efficient campaigns that may close knowledge gaps (*Ariño, Chavan & Otegui, 2016*). It might thus be preferable, and accrue more information, to sample those cells were rare species are distributed, where records are old, and where changes in land use have occurred since the locality was last sampled. This would improve survey designs and efficient monitoring of the wildlife, returning maximum information while spending minimum sampling efforts.

However, as we have only considered just one factor affecting obsolescence of the data—land use changes—our results should be interpreted with caution. Other elements should be explored that might be likely playing a role in the obsolescence of data. For example, changes in land uses can affect species differently according to their life histories

(*De Palma et al., 2015*) and, therefore, the biology of a species could likely be a factor affecting obsolescence.

In this study we have explore the obsolescence of biodiversity records related to the land uses changes. We have pointed out that these records can be outdated and might give us an out-of-time picture of species distribution. However, we have only scratched the surface and we still have to explore in depth the implications of obsolescence of data and whether the use of obsolete data can be misleading towards understanding current species distribution patterns.

## CONCLUSION

As time goes by, we can expect to observe a steady increase in the availability and use of biological records but by definition, records taken at a fixed point in time will age: their validity or fitness-for-use may decrease as the gap between their *recording* time and their *application* time increases. Continuing digitization and release of data from museums or private collections, which may likely include old records, will increase the need to determine, and account for, obsolescence of those records as its age could compromise the knowledge and perception of current distributional patterns of biodiversity worldwide.

In geographical terms, we have tested a relatively local dataset. However, we believe that our observations could be generalized to other cases and other extents. Land uses changes have been recorded from regional to global scales (*Feranec et al., 2010*; *Ellis et al., 2010*) and databases seem to share the same limitations of spatial, taxonomical and temporal biases (*Sastre & Lobo, 2009*; *Boakes et al., 2010*; *Isaac & Pocock, 2015*). Although we could not find literature to compare our results to other similar analyses, we can expect that our findings within the MZNA database in Navarra can be reproduced in other databases as well.

## ACKNOWLEDGEMENTS

We are very grateful to Dr Carmen Escala who started to document the small mammals' community of Navarra in the department of Environmental Biology of University of Navarra, actively contributing to the creation of the 'Pellet sampling' dataset from the MZNA. Without her effort and dedication of inventorying the small mammals' community, we could not have done this research. We also thank all researchers, technicians, and volunteers that contributed with information to the dataset. Finally, we would like to thank Eva Escribano for language comments and two anonymous reviewers for valuable comments that considerably improved the manuscript.

### Funding

NE is funded by Asociación de Amigos de la Universidad de Navarra. The funders had no role in study design, data collection and analysis, decision to publish, or preparation of the manuscript.

## Grant Disclosures

The following grant information was disclosed by the authors:

NE is funded by Asociación de Amigos de la Universidad de Navarra.

## Competing Interests

The authors declare there are no competing interests.

## Author Contributions

- Nora Escribano conceived and designed the experiments, performed the experiments, analyzed the data, wrote the paper, prepared figures and/or tables.
- Arturo H. Ariño conceived and designed the experiments, contributed reagents/materials/analysis tools, reviewed drafts of the paper.
- David Galicia conceived and designed the experiments, performed the experiments, analyzed the data, contributed reagents/materials/analysis tools, reviewed drafts of the paper.

## Data Availability

GBIF

DOI: 10.15470/qomfu6

http://www.gbif.org/dataset/95ed1fa5-2923-4459-836b-11ad8cc4bf42.

## Supplemental Information

Supplemental information for this article can be found online at http://dx.doi.org/10.7717/peerj.2743#supplemental-information.

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
