# Peer review of "Biodiversity data obsolescence and land uses changes"

_PeerJ, doi:10.7717/peerj.2743_

## Round 0.1 · original submission · Major Revisions

Both my reviewers and I think this is an important and timely study. Please do your best to attend to Reviewer 2's rather more extensive comments as thoroughly and quickly as you can.

Reviewer 1 ·

Basic reporting

The reporting is fine, although the paper would greatly benefit from another round or two of editing for English language consistency. There are several areas of the paper where lack of consistency leads to confusion around key results (e.g. first sentence of results section in abstract).

Experimental design

the experimental design seems ok, if a bit overblown for what the authors are trying to show. I detail this further in comments to authors.

Validity of the findings

The findings are extremely simplistic, and the validity of the findings I'm sure holds insofar as they prove a more general point about the lack of appreciation for how out of data observations can lead to mismatches with what analysis someone is trying to do.

Additional comments

I think this manuscript is an important one in that it makes a point rarely appreciated in the species distribution modelling community - which is the age of a biodiversity observation and its potential mismatch with the covariates it is combined with in any statistical model. However, I would argue that the authors take an incredible amount of time to explain, describe, etc. this important, but simple, point. The abstract pretty much says it all. Going much beyond the simple observation that many observations are old and the landscape has changed since they were collected is a bit overkill, as the additional insights that the extended analysis brings is only narrowly confined to the landscape the authors analyze.

Moreover, it seems less important to generally worry about the 'obsolescence' (a rather tortured word anyway) as evidenced simply by age, and more whether or not any input features/covariates of a model match BOTH the spatial AND temporal resolution of the observation data. If one were doing a simple model of observation versus elevation then age of the observation would have little to say about obsolescence. If thinking about landcover then it is obviously important. One would think a more compelling argument would be to show how some standard species distribution modelling outputs would fundamentally change depending on whether each covariate was spatially and temporally matched to the biodiversity observation or not.

Finally, it seems the owl example is a funny one, as the scale at which the errors exist for owl pellet analysis is probably greater than the scale at which the analysis is run.

As you can see, I think the overall point made in the article is a solid and simple one. I worry about the unnecessary complexity of how the authors have approached making their point and feel they have perhaps missed the more general observation that each biodiversity observation should be matched with covariates that exists at the correct spatial and temporal scale.

Reviewer 2 ·

Basic reporting

In general, I think this is a valuable and important line of inquiry that the authors are exploring. I have not previously seen the issue of temporal obsolescence explored in a quantitative way. There are some substantial concerns with the current manuscript, however.

The English grammar could be substantially improved. The discussion section in particular has many errors in the writing. It is poorly structured and difficult to follow. This is detracting from the findings of the research.

Lines 66-70 – I do not think it is correct to say that aquatic systems and oceans have more standardized data. The UK has many regular monitoring programs, from birds to butterflies, as do Australia and South Africa, and probably other countries. The Breeding Bird Survey in the U.S. has been going for 50 years. In general, I believe the authors need to better characterize what data already do exist.

Line 79 – what is meant by a tuple?

Experimental design

There are datasets other than GBIF, and for some countries GBIF holds relatively little of the PBR data. This is not specifically a problem for this study, but it is worth mentioning.

Line 152 – A spatial resolution of 25km2 is extremely coarse. It is inconsistent with the use of 1:5,000 aerial photos or satellite imagery. If this were the resolution of the input land use maps, the later analyses at the same resolution do not make sense.

In general I do not fully understand the land use change methods, even after reading the text several times. Either the methods do not make sense, or the description of them is not clear. Either way, this is a critical concern.

The analyses would be much stronger if the change analysis was done per point rather than generalized to grid cells. Artificially binning things into grid cells of an arbitrary size means that some points will fall in the middle of cells but others may be close to the border between two cells. Consequently, where a point happens to fall within a cell affects how representative the value of that cell actually is for a given point. What would be more informative would be an assessment of change within a set radius of the locality point. Depending on the format and design of the landuse data, which is not clear in the current text, such an evaluation should be relatively straightforward with ArcGIS.

Validity of the findings

I do not doubt the overall conclusion that many biodiversity records are outdated and not representative of what occurs in a location today. It is important and people should be more considerate of this limitation of the PBR data. However, it is not clear to me that the approach used for measuring those changes, specifically the resolution of the land use change mapping, is appropriate or clearly explained.

---

## Round 0.2 · accepted · Accept

I am happy with the revisions that you have made and I see no need to send this for further review. It's a most useful contribution.

As you saw, my reviewers raised issues about your writing on the first version. It's much better, but it could be better still. Since PeerJ does not copy-edit, what you submit is what gets published. To that end, you might want to revise the paper having run it through a programme such as Grammarly (www.grammarly.com). Even after 300 papers, I find this programme helpful and it certainly helps my students. It appears to be free in some places, though I pay for it. It's worth it.

Now, if you chose to make such corrections, please let the editor at PeerJ know ** immediately** — and get the changes back to us quickly. I have not asked for "minor revisions" since if you are happy with the text then it should go to publication as is. I leave this up to you.